# Physical exercise and academic performance among students in the medical field at Jordanian Universities

Ahlam J. Alhemedi[1]*, Sawsan Abuhammad[2,3], Thekraiat Majed A.L. Quran[1], Omar Khasawneh[1], Motaz Al-Yafeai[1], Mohammed Al-Wazeer[1]

**1** Department of Public Health and Family Medicine, Faculty of Medicine, Jordan University of Science and Technology, Irbid, Jordan, **2** Department of Maternal Child Health and Midwifery, Faculty of Nursing, Jordan University of Science and Technology, Irbid, Jordan, **3** Department of Nursing, College of Health Sciences, University of Sharjah, Sharjah, United Arab Emirates

* ajalhmedy@just.edu.jo

## Abstract

### Background

The literature emphasizes multiple factors that affect academic performance, including emotional and physical well-being, motivation, and physical exercise. University students, especially medical students, usually lack time and experience high levels of stress, leading to neglect of a healthy lifestyle. This study aimed to assess the association between physical activity patterns and academic performance (cumulative grade point average (cGPA)) among medical students at Jordanian universities.

### Methods

This is an online survey study that was conducted in Jordan between 28/12/2023 and 30/07/2025. A multivariate logistic regression analysis was performed to identify sociodemographic and physical activity items associated with academic performance, while sociodemographic characteristics were analyzed to predict the probability of engaging in physical activities among participating students.

### Results

A total of 1,209 university students participated in this study. Around 54.8% reported engaging in physical activity. Vigorous exercises such as running or football were practiced by 61.8%, and moderate activities such as brisk walking or swimming by 74.8%. Nearly half of the participants (45.9%) had been exercising for over a year, and 42.7% exercised more than 2.5 hours weekly. The main motivations for physical activity included maintaining health (39.9%) and stress relief (31.8%), while barriers included a lack of time (33.1%) and energy (26.9%). Students aged 22–24 were significantly less likely to have cGPA>3.5 compared to those aged 18–21 (adjusted

**Data availability statement:** All data are available on the following public repository: Figshare: 10.6084/m9.figshare.30608345.

**Funding:** This study was supported by Jordan University of Science and Technology (research grant no: 20240008). The funders had no role in study design, data collection and analysis, decision to publish, or preparation of the manuscript.

**Competing interests:** The authors have declared that no competing interests exist.

odds ratio (aOR) = 0.6, 95% confidence interval (CI): 0.4–0.9; p = 0.021). Additionally, non-Jordanian students had a lower likelihood of achieving a cGPA > 3.5 compared to Jordanian students (aOR = 0.3; 95% CI: 0.2–0.5; p < 0.001).

## Conclusion

In this study sample, medical students exercise moderately. Health improvement and stress relief were the main motivators, whereas a lack of time and energy were barriers. Significant cGPA variances were seen according to age, university, and nationality, revealing sociocultural and institutional impacts within this study population. Although physical activity was not significantly associated with academic performance, females showed lower odds of participating in physical activity, highlighting the importance of promoting physical activity among women to improve overall student health and well-being.

## Introduction

The academic performance of medical students is the principal objective of medical education. High educational achievement is linked with better career opportunities and professional success [1–3]. In contrast, academic failure is associated with several difficulties, including low learning and education quality, as well as an increased unemployment rate [4]. As such, determining the factors that influence academic performance is important [5]. Literature emphasizes multiple factors that affect academic performance [6], including emotional and physical well-being, motivation [7], and physical exercise [8–10]. Research has shown that physical exercise can enhance academic performance by improving cognitive functions, such as memory, concentration, perception, and decision-making [8–11]. Additionally, it is also associated with a better quality of life, enhanced mental health, and is a fundamental factor in preventing considerable diseases [12,13]. It is important to note, exercise's impact on academic performance is affected by frequency, duration, and type [14,15].

University students, especially medical students [16–20], usually lack time and experience high levels of stress, leading them to neglect a healthy lifestyle, including not participating in regular physical activity [21]. Moreover, medical students experience multiple mental health problems, including stress, depression, and anxiety [22–27]. In addition, sedentary behavior is linked to an increased risk of chronic diseases (depression, obesity, cancer, type 2 diabetes, and heart disease) and a higher mortality rate [28]. Thus, a healthy lifestyle (sleep, diet, and exercise) is critical to preserving medical students' well-being [22]. A previous study showed that medical students and physicians who follow a healthy lifestyle are more effective in discussing healthy lifestyle choices with patients, and patients trust their recommendations more [29].

Physical activity offers multiple benefits for medical students, including improving their quality of life and academic performance [30], as well as decreasing stress, burnout, and depression [17,31–34]. Research has also reported that academic performance

and engagement in physical activity are influenced by students' body mass index, age, and gender [35,36]. Understanding the association between academic performance and physical exercise is complex and highly significant. The bulk of earlier investigations demonstrated a positive association between academic performance and levels of physical activity [37–42]; however, few investigations revealed a negative association [43,44] or a non-significant association [30,45]. Despite the global evidence linking physical activity and academic performance, few studies have explored this association in the Middle East or amongst Jordanian students, where sociocultural norms and academic stress may affect exercise habits. Understanding this relationship among Jordanian medical students is essential for the development of academic programs. This study aimed to assess the association between physical activity and academic performance among medical students at Jordanian universities. We hypothesized that students who engage more frequently in physical exercise would demonstrate higher academic achievement.

## Methods

### Study design

This is an online survey study that was conducted in Jordan between 28/12/2023 and 30/07/2025. Data collection and recruitment occurred across three main time windows to increase coverage of medical students during different academic periods. The first wave was conducted between December 2023 and February 2024, which was the mid-academic semester, the second wave was conducted between May and July 2024, which was the final exam period, and the last wave was conducted between March and the end of July 2025 to include the following semester and summer semester students. Recruitment across these time windows was intended to include students from different phases to reduce the potential risk of response bias due to exams.

### Study population and sampling strategy

The study population consisted of active university students currently enrolled in medical, dental, or PharmD programs in Jordan. The inclusion criteria were students aged 18 years or older who were actively attending classes in a medical-related field. Exclusion criteria included students who were not currently enrolled, were on leave of absence, or did not agree to participate. Students with chronic illness, disabilities or other conditions were excluded from the study as these factors could independently influence their physical activity levels and academic performance. No subgroup analyses were performed due to the small number of participants.

### Survey administration

We employed the convenience sampling technique to recruit study participants who met the inclusion criteria. The survey was distributed online via social media platforms, such as Facebook and WhatsApp. The invitation letter clearly mentioned the study purpose, inclusion criteria, and that students participated voluntarily. To facilitate anonymity, no identifying personal information, such as name, email, or student identification document (ID), was collected, and responses were recorded anonymously. Participants were informed that completing the survey implied consent to participate in the study, which aligned with ethical guidelines for online surveys.

### Questionnaire tool

We developed an anonymous questionnaire to examine the association between physical activity and academic performance among medical students in Jordan based on a previous literature review. The questionnaire tool comprised two sections. The first section consisted of seven items to examine the sociodemographic and academic characteristics of the study participants (gender, age, university name, nationality, speciality, cumulative grade point average (cGPA), and cGPA in symbols), which were assessed using multiple-choice and categorical response options. The second section included 10 items on physical activity patterns, frequency, duration, types of exercise, motivations and barriers, using a

combination of multiple-choice and yes/no format questions. The sports injury question referred to lifetime history (ever having had a sports-related injury).

The initial survey instrument was pilot tested on a sample of 25 participants. Piloting the instrument confirmed the clarity of the questionnaire tool with no suggested modifications.

### Ethical approval

Ethical approval for this study was obtained from the Institutional Review Board at Jordan University of Science and Technology, Irbid, Jordan (No: 807/2023). Participants were informed that completing the questionnaire was considered informed consent.

### Data analysis

Data was analyzed using IBM Statistical Package for Social Science (SPSS) Statistics version 31. Descriptive statistics such as frequency and percentage were utilized to summarize categorical data such as demographic characteristics and other physical activity items. The primary outcome was high academic performance, defined as cGPA>3.5. This cutoff point was determined based on the university's grading system, with greater than 3.5 representing a distinguished cGPA. A multivariate logistic regression was conducted to examine factors associated with high academic performance, including sociodemographic characteristics, such as age, gender, nationality, and university, as well as physical activity data, such as frequency and duration of activity. Physical activity was treated as a key predictor alongside demographic covariates. Adjusted odds ratio (aOR) with 95% confidence interval (CI) and p-values were reported. All statistical analyses were two-tailed, conducted at a significance level of $p < 0.05$.

### Results

A total of 1,209 university students participated in the study. 537 (44.4%) were males and 672 (55.6%) were females. Most participants were aged 22–24 years, totaling 818 students (67.7%). Most students were from the Jordan University of Science and Technology (n = 1,012, 83.7%). The majority were Jordanian nationals (n = 1,008, 83.4%), and most were enrolled in the medicine program (n = 1,078, 89.2%). Further details about sociodemographic and academic characteristics are provided in Table 1.

Of 1,209 university students, 663 (54.8%) reported engaging in physical activity. 410 (61.8%) participants did vigorous exercises, such as running or football, and 496 (74.8%) reported doing moderate activities such as brisk walking or swimming. Nearly half (45.9%) had been exercising for over a year, and 283 (42.7%) exercised more than 2.5 hours weekly. The main motivations for doing exercise included maintaining health (39.9%) and stress relief (31.8%), while barriers included a lack of time (33.1%) and energy (26.9%). See Table 2 for more detail.

Students aged 22–24 were significantly less likely to have cGPA>3.5 compared to those aged 18–21 (aOR = 0.6, 95% CI: 0.4–0.9; $p = 0.021$). Compared to Jordan University of Science and Technology students, those from The University of Jordan (aOR= 0.3, 95% CI: 0.1–0.8; $p = 0.018$) and The Hashemite University (aOR = 0.2; 95% CI: 0.1–0.5; $p < 0.001$) were significantly less likely to have a higher cGPA. Additionally, non-Jordanian students were less likely to achieve a cGPA>3.5 compared to Jordanian students (aOR = 0.3; 95% CI: 0.2–0.5; $p < 0.001$). See Table 3.

Female students were significantly less likely to engage in physical activity compared to males (aOR = 0.5; 95% CI: 0.4–0.6; $p < 0.001$). No other variables, including age, university, nationality, specialty, or cGPA, showed a statistically significant association with physical activity in this model. See Table 4.

### Discussion

The results of this study indicate that around 54.8% of medical students engage in physical activity. Vigorous exercises, such as running or football, were performed by 61.8%, and moderate activities such as brisk walking or swimming, were performed by 74.8%. Females were less likely to do physical exercise compared to males. Age group and university of study were factors associated with variation in academic performance.

**Table 1. Sociodemographic and academic characteristics.**

| Sociodemographic and academic characteristics | | N | % |
|---|---|---|---|
| Gender | Male | 537 | 44.4% |
| | Female | 672 | 55.6% |
| Age (years) | 18-21 | 274 | 22.7% |
| | 22-24 | 818 | 67.7% |
| | 25 and older | 117 | 9.7% |
| University | Jordan University of Science and Technology | 1012 | 83.7% |
| | The University of Jordan | 48 | 4.0% |
| | Yarmouk University | 37 | 3.1% |
| | The Hashemite University | 81 | 6.7% |
| | Mutah University | 31 | 2.6% |
| Nationality | Jordanian | 1008 | 83.4% |
| | Other | 201 | 16.6% |
| Specialty | Medicine | 1078 | 89.2% |
| | Dentistry | 67 | 5.5% |
| | PharmD | 64 | 5.3% |
| Cumulative GPA | 4-4.2 | 105 | 8.7% |
| | 3.5-3.99 | 604 | 50.0% |
| | 3-3.49 | 393 | 32.5% |
| | <3 | 107 | 8.9% |
| Cumulative GPA in symbol | A | 610 | 50.5% |
| | B | 480 | 39.7% |
| | C | 119 | 9.8% |

GPA: Grade point average.

Medical students in Jordan face unique institutional and cultural constraints that may contribute to lower physical activity levels and academic performance differences. In our results, only 54.8% reported exercising; this low percentage is mainly due to long study hours and an intense curriculum. Our results also showed that females were less likely to engage in physical activity, likely due to restrictions in the Middle East [46,47]. Students aged 22–24 and non-Jordanians were less likely to achieve high academic performance (cGPA > 3.5), possibly due to a higher academic load and challenges related to adapting to a different environment.

In comparison, prior studies among medical students found that about 81% of the students in Poland [48], 66% in Brazil [49], 62% in India [50,51], 56% in Sudan [52], 53% in Ireland [53], 47% in Saudi Arabia [42], and 32% in Lithuania [54] engage some form of regular physical activity. The observed differences might be attributed to factors related to colleges, including busy curricula, sports facilities, and physical activity programs [55]. For instance, an earlier investigation revealed an association between regular physical activity among medical students, the availability of sports facilities, and the college schedule [48]. Moreover, factors such as students' gender and knowledge also affect the likelihood of engaging in physical activity. The bulk of prior research has highlighted that males tend to engage in physical activity more than females [56], which aligns with our findings. Indeed, regular physical activity is linked with many health, economic, academic, and social benefits [50,57]. Therefore, there is a need to increase participation in regular physical activity among medical students in Jordan. Improving medical students' knowledge about the health benefits of physical activity will be beneficial [48]. In addition, it is essential to take into account specific college and student-related factors when developing any interventions aimed at increasing students' physical activity.

**Table 2. Physical activity patterns, motivations and barriers among university students.**

| Physical activity patterns, motivations and barriers | | N | % |
|---|---|---|---|
| Do you do any type of physical exercise? | Yes | 663 | 54.8% |
| **Do you do any vigorous-intensity sports, fitness or recreational (leisure) activities that cause large increases in breathing or heart rate for at least 10 minutes continuously?** | | | |
| Running or football | Yes | 410 | 61.8% |
| Brisk walking, cycling, swimming, volleyball | Yes | 496 | 74.8% |
| For how long have you been doing regular physical exercise? | < 6 months | 238 | 35.9% |
| | 6 months – 1 year | 121 | 18.3% |
| | > 1 year | 304 | 45.9% |
| Approximately how long do you exercise each week? | 0-2.5 hours (0–150 minutes) weekly | 380 | 57.3% |
| | > 2.5 hours (more than 150 minutes) week | 283 | 42.7% |
| **What types of exercises do you usually do?** | | | |
| Brisk Walking | Yes | 456 | 68.8% |
| Running | Yes | 353 | 53.2% |
| Swimming | Yes | 151 | 22.8% |
| Aerobic exercises | Yes | 298 | 44.9% |
| Football | Yes | 228 | 34.4% |
| Heavy lifting | Yes | 310 | 46.8% |
| How many days per week do you exercise? | Daily | 159 | 24.0% |
| | Every other day | 298 | 44.9% |
| | Twice a week | 129 | 19.5% |
| | Once a week | 77 | 11.6% |
| Where do you usually exercise? | Home | 219 | 33.0% |
| | Gym | 277 | 41.8% |
| | Outdoor | 167 | 25.2% |
| **What is your motivation for doing physical exercises?** | | | |
| Maintaining health | Yes | 482 | 39.9% |
| Leisure | Yes | 129 | 10.7% |
| Stress relief and relaxation | Yes | 385 | 31.8% |
| Physical appearance &body building | Yes | 372 | 30.8% |
| **What are the main reasons preventing you from doing physical exercises?** | | | |
| Lack of time | Yes | 400 | 33.1% |
| Lack of energy | Yes | 325 | 26.9% |
| Lack of resources (money, places, transportation) | Yes | 197 | 16.3% |
| Lack of motivation | Yes | 314 | 26.0% |
| Social influences | Yes | 22 | 1.8% |
| Fear of injury | Yes | 22 | 1.8% |
| Having a chronic illness | Yes | 25 | 2.1% |
| I think it is not important | Yes | 15 | 1.2% |
| Have you ever had a sports injury? | Yes | 344 | 28.5% |
| **What type of injury have you had?** | | | |
| Fracture | Yes | 95 | 7.9% |
| Dislocation | Yes | 51 | 4.2% |
| Sprain (tear of a ligament) | Yes | 142 | 11.7% |
| Strain (tear in tendon or muscle) | Yes | 108 | 8.9% |
| Tendinitis | Yes | 51 | 4.2% |
| Bursitis | Yes | 31 | 2.6% |

**Table 3. Factors associated with higher academic performance.**

| Factors associated with higher academic performance | | aOR (95% CI) | P value |
|---|---|---|---|
| Gender | Male | Reference | |
| | Female | 1.2 (0.8–1.8) | 0.407 |
| Age (years) | 18-21 | Reference | |
| | 22-24 | 0.6 (0.4–0.9) | 0.021 |
| | 25 and older | 0.5 (0.3–1.1) | 0.084 |
| University | Jordan University of Science and Technology | Reference | |
| | The University of Jordan | 0.3 (0.1–0.8) | 0.018 |
| | Yarmouk University | 1.0 (0.4–2.6) | 0.991 |
| | The Hashemite University | 0.2 (0.1–0.5) | < 0.001 |
| | Mutah University | 0.5 (0.2–1.4) | 0.182 |
| Nationality | Jordanian | Reference | |
| | Others | 0.3 (0.2–0.5) | < 0.001 |
| Specialty | Medicine | Reference | |
| | Dentistry | 1.2 (0.5–2.5) | 0.697 |
| | PharmD | 0.9 (0.4–2.1) | 0.805 |
| Do you do any vigorous-intensity sports, fitness or recreational (leisure) activities that cause large increases in breathing or heart rate, such as running or football, for at least 10 minutes continuously? | No | Reference | |
| | Yes | 0.9 (0.6–1.3) | 0.428 |
| Do you do any moderate-intensity sports, fitness or recreational (leisure) activities that cause a small increase in breathing or heart rate, such as brisk walking, cycling, swimming or volleyball, for at least 10 minutes continuously? | No | Reference | |
| | Yes | 1.3 (0.8–1.9) | 0.252 |
| For how long have you been doing regular physical exercise? | < 6 months | Reference | |
| | 6 months – 1 year | 0.9 (0.5–1.4) | 0.567 |
| | > 1 year | 1.3 (0.9–1.9) | 0.235 |
| Approximately how long do you exercise weekly? | 0-2.5 hours (0–150 minutes) weekly | Reference | |
| | > 2.5 hours (more than 150 minutes) week | 1.2 (0.8–1.8) | 0.431 |
| | if you do exercises for more than 150 minutes (>;2.5 hours) weekly; please write down approximately how many hours you exercise per week? | 1.0 (1.0–1.0) | 0.979 |
| How many days per week do you exercise? | Daily | Reference | |
| | Every other day | 1.1 (0.7–1.6) | 0.793 |
| | Twice weekly | 1.2 (0.7–2.0) | 0.534 |
| | Once weekly | 1.0 (0.5–1.8) | 0.884 |
| Where do you usually exercise? | Home | Reference | |
| | Gym | 1.1 (0.7–1.8) | 0.623 |
| | Outdoor | 1.2 (0.7–1.9) | 0.504 |

aOR: adjusted odds ratio.

This study showed that maintaining health (39.9%) and relieving stress (31.8%) were the primary motivations for engaging in physical activity, while a lack of time (33.1%) and energy (26.9%) were the most significant barriers. These findings align with several prior studies among medical students in India, Poland, the Western Balkans, and the United States, which report that improving stamina [51] and well-being [58,59], enhancing physical appearance [51,58,59], promoting mental health [60], reducing stress [59], and losing weight [51] were the most commonly reported motivations for

**Table 4. Factors associated with physical activity engagement.**

| Factors associated with physical activity engagement | | aOR (95% CI) | P value |
|---|---|---|---|
| Gender | Male | Reference | |
| | Female | 0.5 (0.4–0.6) | < 0.001 |
| Age (years) | 18-21 | Reference | |
| | 22-24 | 1.2 (0.9–1.6) | 0.187 |
| | 25 and older | 1.0 (0.6–1.6) | 0.976 |
| University | Jordan University of Science and Technology | Reference | |
| | The University of Jordan | 0.9 (0.5–1.8) | 0.856 |
| | Yarmouk University | 1.6 (0.8–3.2) | 0.189 |
| | The Hashemite University | 1.4 (0.9–2.4) | 0.150 |
| | Mutah University | 0.7 (0.3–1.5) | 0.336 |
| Nationality | Jordanian | Reference | |
| | Other | 1.0 (0.7–1.4) | 0.920 |
| Specialty | Medicine | Reference | |
| | Dentistry | 1.2 (0.7–2.1) | 0.437 |
| | PharmD | 1.0 (0.6–1.7) | 0.920 |
| cGPA | <3.5 | Reference | |
| | ≥ 3.5 | 1.3 (1.0–1.6) | 0.071 |

aOR: adjusted odds ratio; cGPA: cumulative grade point average.

doing physical exercise. Consistent with our findings, the leading barriers identified in earlier investigations were a lack of time [51,58,60,61] and energy [22]. Faculty obligations [59] and excessive study load [52] were also among the most prevalent physical activity barriers among medical students. Understanding these motivations and barriers is crucial, as the literature documents the association between motivation and levels of physical activity [61]. Moreover, low levels of physical activity were linked to poor mental health and academic performance [62]. Consequently, addressing the identified barriers is critical. Research highlighted possible solutions to overcome these barriers including making students' schedule more consistent providing specific exercise breaks [22].

Our study revealed that students aged 22–24 were significantly less likely to have a cGPA > 3.5 compared to those aged 18–21 (aOR = 0.6, 95% CI: 0.4–0.9; p = 0.021). Additionally, compared to Jordan University of Science and Technology students, those from The University of Jordan (aOR = 0.3, 95% CI: 0.1–0.8; p = 0.018) and The Hashemite University (aOR = 0.2; 95% CI: 0.1–0.5; p < 0.001) were significantly less likely to have a higher cGPA. Additionally, non-Jordanian students were less likely to achieve a cGPA > 3.5 compared to Jordanian students (aOR = 0.3; 95% CI: 0.2–0.5; p < 0.001). These findings match those observed in earlier studies. For example, a prior survey among medical students revealed a significant association between age and cGPA; the lower the students' age, the higher their cGPA [63]. Similarly, several previous studies demonstrated significantly better academic performance among younger students than older ones [64–67]. These differences in academic performance may reflect family obligations, economic problems [64,68], stress [69,70], and obstacles to learning and adaptation [71–75] among older students. Thus, targeted academic interventions are necessary for older students.

The literature also supports the observation of a higher cGPA among Jordan University of Science and Technology students compared to other university students, and among Jordanian students compared to non-Jordanian students. Each university has a different admissions process, learning environment, and curriculum, and all these aspects are documented to influence students' academic performance [76–82]. Consistent with this study, international students and medical students in the United Kingdom [83] and New Zealand [84] have poor academic performance as a result of mental

health challenges and concerns about quality of life. Other studies among medical students have shown an association between mental health issues and low educational achievement [85] or low cGPA [62]. Therefore, it is imperative to identify any mental health problems among these students early [83] and manage them effectively. This may lead to improved academic performance among medical students. Other related factors identified in the literature include coping strategies, cultural factors, English language ability, marital status, gender, and age [70,85–89]. As such, it is necessary to consider these factors when developing interventions to enhance medical performance among international students.

Based on the study findings, universities in Jordan ought to provide exercise breaks, improve wellness programs, and promote campaigns related to physical activities, such as activities that reduce stress and improve mental health. This will positively impact the academic performance of students, especially for non-Jordanian students who are suffering from adaptation challenges.

## Conclusion

Overall, the findings of this research emphasize that medical students engage in moderate levels of physical activity. A lack of time and energy were significant barriers in our study sample, while health maintenance and tension relief emerged as the primary motivators. Potential sociocultural and institutional influences were suggested by the significant variation in academic performance based on age, university, and nationality in our study sample. The need for targeted strategies to encourage regular physical activity and support academic achievement across a variety of student groups is suggested by these findings.

## Acknowledgments

None.

## Author contributions

**Conceptualization:** Ahlam J Alhemedi.

**Data curation:** Ahlam J Alhemedi.

**Formal analysis:** Ahlam J Alhemedi.

**Funding acquisition:** Ahlam J Alhemedi.

**Investigation:** Ahlam J Alhemedi, Sawsan Abuhammad, Thekraiat Majed AL Quran, Omar Khasawneh, Motaz Al-Yafeai, Mohammed Al-Wazeer.

**Methodology:** Ahlam J Alhemedi.

**Project administration:** Ahlam J Alhemedi.

**Resources:** Ahlam J Alhemedi, Sawsan Abuhammad, Thekraiat Majed AL Quran, Omar Khasawneh, Motaz Al-Yafeai, Mohammed Al-Wazeer.

**Software:** Ahlam J Alhemedi.

**Supervision:** Ahlam J Alhemedi.

**Validation:** Ahlam J Alhemedi.

**Visualization:** Ahlam J Alhemedi.

**Writing – original draft:** Ahlam J Alhemedi.

**Writing – review & editing:** Ahlam J Alhemedi, Sawsan Abuhammad, Thekraiat Majed AL Quran, Omar Khasawneh, Motaz Al-Yafeai, Mohammed Al-Wazeer.

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
