## [Decision Letter · Decision Letter 0]

5 Nov 2025

Dear Dr, Al-Hemedi,

Thank you for submitting your manuscript to PLOS ONE. After careful consideration, we feel that it has merit but does not fully meet PLOS ONE’s publication criteria as it currently stands. Therefore, we invite you to submit a revised version of the manuscript that addresses the points raised during the review process.

We look forward to receiving your revised manuscript.

Kind regards,

Nour Amin Elsahoryi, pHD

Academic Editor

PLOS ONE

“This study was supported by Jordan University of Science and Technology (research grant no: 20240008).”

“This study was supported by Jordan University of Science and Technology (research grant no: 20240008).”

“This study was supported by Jordan University of Science and Technology (research grant no: 20240008).”

4. In the online submission form you indicate that your data is not available for proprietary reasons and have provided a contact point for accessing this data. Please note that your current contact point is a co-author on this manuscript. According to our Data Policy, the contact point must not be an author on the manuscript and must be an institutional contact, ideally not an individual. Please revise your data statement to a non-author institutional point of contact, such as a data access or ethics committee, and send this to us via return email. Please also include contact information for the third party organization, and please include the full citation of where the data can be found.

Additional Editor Comments:

Below are consolidated comments and recommendations drawn from both reviewers’ reports and the manuscript content:

1. Study Aim and Conceptual Clarity

The abstract and introduction do not clearly articulate whether the study aims to examine associations between physical exercise and academic achievement, or merely to describe prevalence patterns.

→ Revise the abstract to explicitly state the study’s primary research question, hypothesis, and analytic direction (e.g., “to assess the association between physical activity patterns and CGPA among medical students”).

In the introduction (lines 59–90), the rationale is general and repetitive. Clarify what makes this study novel or contextually relevant to Jordan. For example, emphasize the lack of regional data, sociocultural constraints, or institutional policies affecting student lifestyles.

2. Methodological Transparency

Sampling and Period: Specify how respondents were distributed across the research period (December 2023–July 2025). Reviewer 1 noted that this long duration could overlap with exam sessions and vacation periods, potentially biasing participation and activity levels.

→ Add a short table or sentence explaining data collection waves and timing.

Inclusion/Exclusion Criteria: The inclusion criteria are very broad (“active medical student in Jordan”). No exclusion criteria are described. Clarify whether students with chronic illness, disabilities, or athletic status were excluded or analyzed separately.

Survey Administration: The methods section (lines 94–112) needs more detail on how the survey was distributed and how anonymity was maintained. The text currently conflicts (“anonymous questionnaire” vs. “not anonymous”).

→ Explain the mode of distribution (e.g., via WhatsApp, official email lists) and how consent was obtained.

Questionnaire Design: Mention the number of items per section, response scales (e.g., Likert, multiple choice), and whether any items were adapted from validated instruments.

3. Statistical Analysis

Reviewer 2 correctly notes that the analytical framework is unclear. Currently, you report two logistic regression models—one predicting CGPA and another predicting physical activity.

→ You need to justify this dual-model approach or reframe it as a single multivariate model focusing on academic performance as the primary outcome, with physical activity as a key predictor alongside demographics.

Add clear definitions for “high academic performance” (CGPA >3.5) and explain why this cutoff was chosen.

Ensure that all variables included in the logistic regression are described (independent and dependent).

Consider providing adjusted ORs and confidence intervals in one concise summary table, with p-values properly formatted (e.g., p < 0.05).

4. Results Clarification

In Table 1, ensure that percentages add up and that total counts are consistent.

Explain the discrepancy between inclusion criteria (medical field students only) and the 10.8% of respondents from non-medical programs (Dentistry, PharmD). Clarify whether these were included in analysis or excluded later.

Injury data (Table 2) require more clarity — specify whether injuries were recent or lifetime occurrences.

Simplify the results narrative: avoid repeating entire table contents in text; instead, highlight the key significant associations and trends.

5. Discussion and Interpretation

The discussion section (lines 156–244) currently reads as a broad literature summary rather than an interpretation of your data.

→ Begin by restating the main findings quantitatively (e.g., “Only 54.8% engaged in any physical activity, lower than Poland (81%) and Brazil (66%)”).

→ Discuss why Jordanian students might differ — institutional load, cultural context, gender patterns, or infrastructure.

Narrow the discussion to variables that were significantly related to CGPA (age, nationality, university) and propose plausible explanations.

→ For example, explore how institutional academic load or student living conditions might account for lower performance among non-Jordanian students.

Strengthen the final paragraph by linking the results directly to policy or educational recommendations (e.g., structured exercise breaks, on-campus wellness programs).

6. English and Formatting

Overall readability is acceptable, but there are multiple grammatical and syntactic errors that obscure meaning (e.g., “Health improvement and stress relief their drove motivation” → “Health improvement and stress relief were the main motivators”).

→ A thorough language revision is recommended prior to resubmission.

Avoid redundancy in the abstract and discussion. Simplify long sentences and use consistent tense (past for results, present for interpretation).

7. Data Availability and Ethical Statements

Reviewer 1 noted incomplete compliance with PLOS data-sharing requirements.

→ Ensure that the dataset is either uploaded as Supporting Information or deposited in a public repository with an accessible link, not merely “available on request.”

Summary Recommendation

The study addresses an important public health and educational topic but currently lacks sufficient methodological rigor and analytical focus. Substantial revisions are required in the Methods, Statistical Analysis, and Discussion sections to clarify design, strengthen interpretation, and meet PLOS ONE standards.

Reviewers' comments:

Reviewer's Responses to Questions

**Comments to the Author**

1. Is the manuscript technically sound, and do the data support the conclusions?

Reviewer #1: Partly

Reviewer #2: Partly

2. Has the statistical analysis been performed appropriately and rigorously?

Reviewer #1: Yes

Reviewer #2: No

3. Have the authors made all data underlying the findings in their manuscript fully available?

Reviewer #1: No

Reviewer #2: Yes

4. Is the manuscript presented in an intelligible fashion and written in standard English?

Reviewer #1: Yes

Reviewer #2: No

Reviewer #1: To increase the scientific quality of the article and provide a clearer understanding of the research, it would be desirable to:

1. describe more precisely how the number of respondents is distributed in the selected research period,

2. emphasize whether the stages of student exam sessions and breaks were taken into account,

3. highlight and describe in more detail the reflection of the established correlations in the work.

Reviewer #2: Abstract and Introduction:

The abstract does not clearly demonstrate the relationship between physical exercise and academic achievement. It is unclear whether the study aims to examine an association between the two variables or to report prevalence data. The research objective should be explicitly stated to reflect the study design.

Novelty and Rationale:

The manuscript would benefit from clarifying what makes this study distinctive. Are there contextual or cultural factors in Jordanian universities that could make the findings novel or relevant to local practice? Highlighting such aspects would strengthen the paper’s contribution.

Inclusion and Exclusion Criteria:

The inclusion criteria appear too broad, and no exclusion criteria are described. Consider clarifying whether specific groups such as student athletes or those with medical conditions or disabilities were excluded, as these could influence physical activity and academic outcomes.

Methods:

The data collection process requires more detail. Indicate how the questionnaire was distributed (e.g., through university email lists, WhatsApp groups, or classroom links). The manuscript states that the questionnaire was not anonymous, yet the data were collected anonymously — this needs clarification. Specify:

- The total number of questionnaires distributed and returned.

- The type of questions used (e.g., Likert scale, binary, open-ended).

- How anonymity was maintained and identifiers were handled.

- How participants were informed about and provided informed consent.

Statistical Analysis:

The description of the analysis is confusing. If the primary aim is to examine the relationship between physical exercise and academic performance, a single multivariate logistic regression should be used, with academic performance as the dependent variable and physical exercise as the key predictor. Using two separate logistic regressions may be conceptually inappropriate unless justified.

Results:

The inclusion criteria specify medical students, yet only 89% of respondents were from medical programmes — this discrepancy should be explained. Additionally, the presentation of results lacks clarity regarding injury data (e.g., whether injuries were recent or past). The response options for reasons to exercise or not exercise also appear limited; consider allowing open-ended responses to capture other possible factors.

Discussion:

The discussion is too general and lacks depth. It would be stronger if focused on the relationship between physical activity level and academic achievement rather than addressing too many unrelated variables. A more critical and analytical interpretation of findings in relation to existing literature is recommended.

**Do you want your identity to be public for this peer review?** For information about this choice, including consent withdrawal, please see our Privacy Policy

Reviewer #1: No

Reviewer #2: No

---

## [Author Response · Author response to Decision Letter 1]

15 Nov 2025

Response to editor and reviewers:

Response: Thank you, we have now addressed this point.

“This study was supported by Jordan University of Science and Technology (research grant no: 20240008).”

Response: Thank you, we have now addressed this point.

Response: Thank you, we have now addressed this point.

“This study was supported by Jordan University of Science and Technology (research grant no: 20240008).”

“This study was supported by Jordan University of Science and Technology (research grant no: 20240008).”

Response: We thank the reviewer for this clarification. We have now included the updated Funding and Role of Funder statement in the cover letter as requested.

4. In the online submission form you indicate that your data is not available for proprietary reasons and have provided a contact point for accessing this data. Please note that your current contact point is a co-author on this manuscript. According to our Data Policy, the contact point must not be an author on the manuscript and must be an institutional contact, ideally not an individual. Please revise your data statement to a non-author institutional point of contact, such as a data access or ethics committee, and send this to us via return email. Please also include contact information for the third party organization, and please include the full citation of where the data can be found.

Response: Thank you for this comment, we have now uploaded the data on public repository based on the editor’s comment.

Response: Thank you for this comment, we have now addressed this point.

Response: Thank you for the clarification. We have removed the Ethics Approval statement from the end of the manuscript and incorporated it into the Methods section.

Additional Editor Comments:

Below are consolidated comments and recommendations drawn from both reviewers’ reports and the manuscript content:

1. Study Aim and Conceptual Clarity

The abstract and introduction do not clearly articulate whether the study aims to examine associations between physical exercise and academic achievement, or merely to describe prevalence patterns.

→ Revise the abstract to explicitly state the study’s primary research question, hypothesis, and analytic direction (e.g., “to assess the association between physical activity patterns and CGPA among medical students”).

In the introduction (lines 59–90), the rationale is general and repetitive. Clarify what makes this study novel or contextually relevant to Jordan. For example, emphasize the lack of regional data, sociocultural constraints, or institutional policies affecting student lifestyles.

Response: We thank the reviewer for this helpful comment. The abstract and introduction have been revised to clearly state the study’s aim, analytic direction, and hypothesis. The study aim is now explicitly defined as “to assess the association between physical activity patterns and academic performance (CGPA) among medical students at Jordanian universities.” Additionally, the introduction was modified to emphasize the novelty and contextual relevance of the study by highlighting the lack of regional data and the potential influence of sociocultural and institutional factors on students’ physical activity in Jordan (line 92-101).

2. Methodological Transparency

Sampling and Period: Specify how respondents were distributed across the research period (December 2023–July 2025). Reviewer 1 noted that this long duration could overlap with exam sessions and vacation periods, potentially biasing participation and activity levels.

→ Add a short table or sentence explaining data collection waves and timing.

Response: We thank the reviewer for this valuable suggestion. We have now clarified the distribution of data collection across the study period. Specifically, data were collected in three waves (December 2023–February 2024, May–July 2024, and March–July 2025) to include students during both regular semesters and examination periods, thereby ensuring a representative sample and minimizing potential bias due to academic scheduling (Line 105-111).

Inclusion/Exclusion Criteria: The inclusion criteria are very broad (“active medical student in Jordan”). No exclusion criteria are described. Clarify whether students with chronic illness, disabilities, or athletic status were excluded or analyzed separately.

Response: We thank the reviewer for this comment. We have clarified the inclusion and exclusion criteria in the Methods section. Students with chronic illnesses, disabilities, or professional athletic status were excluded from the study as these factors could independently influence their physical activity levels and academic performance. No subgroup analyses were conducted for these categories due to limited numbers. Exclusion criteria now specify that students not actively enrolled or not consenting to participate were excluded (Line 115-122).

Survey Administration: The methods section (lines 94–112) needs more detail on how the survey was distributed and how anonymity was maintained. The text currently conflicts (“anonymous questionnaire” vs. “not anonymous”).

→ Explain the mode of distribution (e.g., via WhatsApp, official email lists) and how consent was obtained.

Response: We thank the reviewer for this important comment. We have clarified that the survey was distributed online via WhatsApp and Facebook, that participation was voluntary, and that completing the survey implied consent. We also confirmed that the questionnaire was fully anonymous, with no personal identifiers collected, to ensure confidentiality. These details have been added to the Methods section under “Survey Administration”, see lines 123-133.

Questionnaire Design: Mention the number of items per section, response scales (e.g., Likert, multiple choice), and whether any items were adapted from validated instruments.

Response: We thank the reviewer for this comment. We have updated the Methods section to provide a detailed description of the questionnaire, including the number of items per section, types of response scales used (yes/no, multiple-choice, and Likert), and the fact that some items were adapted from validated instruments. This information has been added to the “Questionnaire tool” subsection (lines 135-144). No items were adapted from previous literature. We mentioned that we developed an anonymous questionnaire to examine the association between physical activity and academic performance, see lines 135-136.

3. Statistical Analysis

Reviewer 2 correctly notes that the analytical framework is unclear. Currently, you report two logistic regression models—one predicting CGPA and another predicting physical activity.

→ You need to justify this dual-model approach or reframe it as a single multivariate model focusing on academic performance as the primary outcome, with physical activity as a key predictor alongside demographics.

Add clear definitions for “high academic performance” (CGPA >3.5) and explain why this cutoff was chosen.

Ensure that all variables included in the logistic regression are described (independent and dependent).

Consider providing adjusted ORs and confidence intervals in one concise summary table, with p-values properly formatted (e.g., p < 0.05).

Response: We thank the reviewer for this comment. We have clarified that the primary outcome of interest is high academic performance (CGPA > 3.5), and physical activity is included as a key predictor alongside sociodemographic covariates in a multivariate logistic regression model. We also conducted a secondary logistic regression to explore factors associated with physical activity engagement, see lines 154-163. All independent and dependent variables are now clearly described, and adjusted ORs with 95% CIs are presented in Tables 3 and 4.

4. Results Clarification

In Table 1, ensure that percentages add up and that total counts are consistent.

Explain the discrepancy between inclusion criteria (medical field students only) and the 10.8% of respondents from non-medical programs (Dentistry, PharmD). Clarify whether these were included in analysis or excluded later.

Response: Thank you for your comment. As clarified in the Methods section, the study population consisted of active university students currently enrolled in medical, dental, or PharmD programs (Line 116), which we defined as “medical field” students. Therefore, students from Dentistry (5.5%) and PharmD (5.3%) programs were intentionally included in the analysis. The total number of participants is 1,209, and all analyses were conducted on this complete sample. We have corrected the manuscript to reflect this number consistently throughout the text and tables (Line 167).

Injury data (Table 2) require more clarity — specify whether injuries were recent or lifetime occurrences.

Simplify the results narrative: avoid repeating entire table contents in text; instead, highlight the key significant associations and trends.

Response: We thank the reviewer for the suggestion. “The injury data refer to lifetime occurrences, as participants were asked whether they had ever experienced a sports-related injury. In the results section, we have summarized the key findings rather than repeating the full table to avoid redundancy.”

5. Discussion and Interpretation

The discussion section (lines 156–244) currently reads as a broad literature summary rather than an interpretation of your data.

→ Begin by restating the main findings quantitatively (e.g., “Only 54.8% engaged in any physical activity, lower than Poland (81%) and Brazil (66%)”).

Response: We thank the reviewer for the comment. We have revised the opening paragraph of the discussion to begin with a clear summary of our main findings (Line 198).

→ Discuss why Jordanian students might differ — institutional load, cultural context, gender patterns, or infrastructure.

Response: We thank the reviewer for this suggestion. We have revised the discussion to focus on contextualizing our findings among Jordanian medical students. A new paragraph has been added highlighting potential explanations for lower physical activity and variations in CGPA, including institutional load, sociocultural norms, gender patterns (lines 203-209)

Narrow the discussion to variables that were significantly related to CGPA (age, nationality, university) and propose plausible explanations.

→ For example, explore how institutional academic load or student living conditions might account for lower performance among non-Jordanian students.

Response: We thank the reviewer for this comment; unnecessary paragraphs were deleted to avoid redundancy and improve readability.

Strengthen the final paragraph by linking the results directly to policy or educational recommendations (e.g., structured exercise breaks, on-campus wellness programs).

Response: We thank the reviewer for the suggestion. We have added a concluding paragraph in the discussion section linking our findings to practical policy and educational recommendations

6. English and Formatting

Overall readability is acceptable, but there are multiple grammatical and syntactic errors that obscure meaning (e.g., “Health improvement and stress relief their drove motivation” → “Health improvement and stress relief were the main motivators”).

→ A thorough language revision is recommended prior to resubmission.

Response: We thank the reviewer for this comment, we have now addressed this point.

Avoid redundancy in the abstract and discussion. Simplify long sentences and use consistent tense (past for results, present for interpretation).

Response: We thank the reviewer for this comment, we have now addressed this point as highlighted above.

7. Data Availability and Ethical Statements

Reviewer 1 noted incomplete compliance with PLOS data-sharing requirements.

→ Ensure that the dataset is either uploaded as Supporting Information or deposited in a public repository with an accessible link, not merely “available on request.”

Response: We thank the reviewer for this comment, we have now addressed this point as highlighted above.

Summary Recommendation

The study addresses an important public health and educational topic but currently lacks sufficient methodological rigor and analytical focus. Substantial revisions are required in the Methods, Statistical Analysis, and Discussion sections to clarify design, strengthen interpretation, and meet PLOS ONE standards.

Reviewers' comments:

Reviewer's Responses to Questions

Comments to the Author

1. Is the manuscript technically sound, and do the data support the conclusions?

Reviewer #1: Partly

Reviewer #2: Partly

2. Has the statistical analysis been performed appropriately and rigorously?

Reviewer #1: Yes

Reviewer #2: No

3. Have the authors made all data underlying the findings in their manuscript fully available?

The PLOS Data policy requires authors to make all data underlying the findings described in their manuscript fully available without restriction, with rare exception (please refer to the Data Availability Statement in the manuscript PDF file). The data should be provided as part of the manuscript or its supporting information, or deposited to a public repository. For example, in addition to summary statistics, the data points behind means, medians and variance measures sh

---

## [Editor Report · Decision Letter 1]

8 Dec 2025

Dear Dr.  Al-Hemedi,

Thank you for submitting your manuscript to PLOS ONE. After careful consideration, we feel that it has merit but does not fully meet PLOS ONE’s publication criteria as it currently stands. Therefore, we invite you to submit a revised version of the manuscript that addresses the points raised during the review process.

Please submit your revised manuscript by Jan 22 2026 11:59PM. If you will need more time than this to complete your revisions, please reply to this message or contact the journal office at plosone@plos.org . A rebuttal letter that responds to each point raised by the academic editor and reviewer(s). You should upload this letter as a separate file labeled 'Response to Reviewers'.A marked-up copy of your manuscript that highlights changes made to the original version. You should upload this as a separate file labeled 'Revised Manuscript with Track Changes'.An unmarked version of your revised paper without tracked changes. You should upload this as a separate file labeled 'Manuscript'.

We look forward to receiving your revised manuscript.

Kind regards,

Nour Amin Elsahoryi, pHD

Academic Editor

PLOS One

Journal Requirements:

Additional Editor Comments:

Thanks, your revision was very thorough and your answers detailed. The paper has become a much better job. The primary objective and analytic focus is now well defined, the sampling frame and administration of survey is better outlined, multivariate logistic regression application is more evident and the data is released in an appropriate way in a public repository. As well, you have refocused the Discussion to put your key findings into the Jordanian context, such as the non-relation between physical activity and academic performance and gender difference in physical activity participation.

Prior to acceptance of the paper, I would wish you consider the following minor issues:

Harmonize the sample size throughout the manuscript.

In several places the total sample is reported as 1,209, while one sentence still reads “1,290 209 university students”, which appears to be a residual editing artifact. Please ensure that the same total N is used consistently in the Abstract, Methods, Results, tables, and cover letter.

Thorough language and style check.

The English is generally understandable, but several typographical and grammatical errors remain and should be corrected (e.g. “psychical activity” → “physical activity”, “Health improvement and stress relief their drove motivation” → “Health improvement and stress relief were the main motivators”, “factosfactors” → “factors”, “was utilized” → “were utilized”, consistent use of “cGPA/CGPA”, etc.). A careful proof-reading or professional language editing is recommended to meet journal standards.

Clarify questionnaire development.

In your response letter you state that some items were adapted from validated instruments but also that “No items were adapted from previous literature”, which is contradictory. Please make the Methods section explicit and internally consistent: either

(a) specify which items were adapted and cite the original sources, or

(b) state clearly that all items were developed de novo for this study based on the literature, and remove any mention of adaptation.

Minor clarifications.

Explicitly state in the Methods that the sports-injury question referred to lifetime history (ever having had a sports-related injury), to match the explanation given in your response.

Double-check all tables (especially Tables 2–4) for formatting, consistent decimal places, and p-value notation (e.g. use “p < 0.001” rather than “0.000”).

---

## [Author Response · Author response to Decision Letter 2]

20 Dec 2025

Response to reviewer:

Thanks, your revision was very thorough and your answers detailed. The paper has become a much better job. The primary objective and analytic focus is now well defined, the sampling frame and administration of survey is better outlined, multivariate logistic regression application is more evident and the data is released in an appropriate way in a public repository. As well, you have refocused the Discussion to put your key findings into the Jordanian context, such as the non-relation between physical activity and academic performance and gender difference in physical activity participation.

Prior to acceptance of the paper, I would wish you consider the following minor issues:

Harmonize the sample size throughout the manuscript.

In several places the total sample is reported as 1,209, while one sentence still reads “1,290 209 university students”, which appears to be a residual editing artifact. Please ensure that the same total N is used consistently in the Abstract, Methods, Results, tables, and cover letter.

- Response: We have now addressed this comment.

Thorough language and style check.

The English is generally understandable, but several typographical and grammatical errors remain and should be corrected (e.g. “psychical activity” → “physical activity”, “Health improvement and stress relief their drove motivation” → “Health improvement and stress relief were the main motivators”, “factosfactors” → “factors”, “was utilized” → “were utilized”, consistent use of “cGPA/CGPA”, etc.). A careful proof-reading or professional language editing is recommended to meet journal standards.

- Response: We have now addressed this comment and conducted full proofreading for the manuscript.

Clarify questionnaire development.

In your response letter you state that some items were adapted from validated instruments but also that “No items were adapted from previous literature”, which is contradictory. Please make the Methods section explicit and internally consistent: either

(a) specify which items were adapted and cite the original sources, or

(b) state clearly that all items were developed de novo for this study based on the literature, and remove any mention of adaptation.

- Response: We have now clarified that we developed an anonymous questionnaire to examine the association between physical activity and academic performance among medical students in Jordan based on previous literature review.

Minor clarifications.

Explicitly state in the Methods that the sports-injury question referred to lifetime history (ever having had a sports-related injury), to match the explanation given in your response.

- Response: We have now addressed this comment, see lines 133-134.

Double-check all tables (especially Tables 2–4) for formatting, consistent decimal places, and p-value notation (e.g. use “p < 0.001” rather than “0.000”).

- Response: We have now addressed this comment.

---

## [Editor Report · Decision Letter 2]

1 Jan 2026

Dear Dr. Al-Hemedi,

Thank you for submitting your manuscript to PLOS ONE. After careful consideration, we feel that it has merit but does not fully meet PLOS ONE’s publication criteria as it currently stands. Therefore, we invite you to submit a revised version of the manuscript that addresses the points raised during the review process.

Please submit your revised manuscript within Feb 15 2026 11:59PM. If you will need more time than this to complete your revisions, please reply to this message or contact the journal office at plosone@plos.org . A letter that responds to each point raised by the academic editor and reviewer(s). You should upload this letter as a separate file labeled 'Response to Reviewers'.A marked-up copy of your manuscript that highlights changes made to the original version. You should upload this as a separate file labeled 'Revised Manuscript with Track Changes'.An unmarked version of your revised paper without tracked changes. You should upload this as a separate file labeled 'Manuscript'.

We look forward to receiving your revised manuscript.

Kind regards,

Nour Amin Elsahoryi, pHD

Academic Editor

PLOS One

Journal Requirements:

Additional Editor Comments :

Comments to the Author

Thank you for submitting the revised version (R2) and your detailed response. The manuscript is substantially improved in clarity and structure. In particular, the study objective is now more explicit, the analytical strategy is better organized, and the reporting is generally more consistent. I also note that the previously observed inconsistency in the reported proportion for “moderate” physical activity appears to have been corrected and is now aligned across the abstract/text/tables. The sample size reporting is also more consistent.

Before the manuscript can be accepted, please address the following minor but important points:

Language and copyediting (required)

The revised manuscript still contains multiple obvious typographical/spacing issues (e.g., merged words, duplicated letters, and formatting artifacts).

Please perform a final professional proofreading pass and ensure consistent formatting throughout (headings, spacing, punctuation, abbreviations).

Consistency between the response letter and Methods

Your response includes a contradiction regarding whether any questionnaire items were “adapted” from previously validated instruments.

Please ensure the response letter is internally consistent and fully aligned with the final Methods text. If items were adapted, specify from which sources; if not, remove any wording that implies adaptation.

Avoid overstatement of representativeness

Given the convenience/online recruitment approach, please avoid wording that implies the sampling strategy “ensures” or “guarantees” representativeness.

Please revise such statements to more neutral language (e.g., recruitment across different time windows/academic periods to increase coverage), and acknowledge this appropriately as a limitation where relevant.

---

## [Author Response · Author response to Decision Letter 3]

3 Jan 2026

Response to reviewer:

Thank you for submitting the revised version (R2) and your detailed response. The manuscript is substantially improved in clarity and structure. In particular, the study objective is now more explicit, the analytical strategy is better organized, and the reporting is generally more consistent. I also note that the previously observed inconsistency in the reported proportion for “moderate” physical activity appears to have been corrected and is now aligned across the abstract/text/tables. The sample size reporting is also more consistent.

Before the manuscript can be accepted, please address the following minor but important points:

Language and copyediting (required)

- Thank you for this comment, the currently submitted manuscript has undergone complete proofreading.

The revised manuscript still contains multiple obvious typographical/spacing issues (e.g., merged words, duplicated letters, and formatting artifacts).

- Thank you for this comment, the currently submitted manuscript has undergone complete proofreading.

Please perform a final professional proofreading pass and ensure consistent formatting throughout (headings, spacing, punctuation, abbreviations).

- Thank you for this comment, the currently submitted manuscript has undergone complete proofreading.

Consistency between the response letter and Methods

Your response includes a contradiction regarding whether any questionnaire items were “adapted” from previously validated instruments.

Please ensure the response letter is internally consistent and fully aligned with the final Methods text. If items were adapted, specify from which sources; if not, remove any wording that implies adaptation.

- Thank you for this comment. We have now removed any wording that implies adaptation, see the method section.

Avoid overstatement of representativeness

Given the convenience/online recruitment approach, please avoid wording that implies the sampling strategy “ensures” or “guarantees” representativeness.

- Thank you for this comment. We have now addressed this comment, see line 121.

Please revise such statements to more neutral language (e.g., recruitment across different time windows/academic periods to increase coverage), and acknowledge this appropriately as a limitation where relevant.

- Thank you for this comment. We have now addressed this comment, see lines 100-107.

---

## [Editor Report · Decision Letter 3]

6 Jan 2026

Physical Exercise and Academic Performance among Students in the Medical field at Jordanian Universities

PONE-D-25-49996R3

Dear Dr. Al-Hemedi,

We’re pleased to inform you that your manuscript has been judged scientifically suitable for publication and will be formally accepted for publication once it meets all outstanding technical requirements.

Kind regards,

Nour Amin Elsahoryi, pHD

Academic Editor

PLOS One

Additional Editor Comments (optional):

Thank you for the revised version—overall, the paper is now acceptable in content and structure. Only minor points remain:

Consistency check: Please ensure the “moderate activity” percentage/value is identical across the manuscript, tables, and submission letters.

Final proofreading: Do one last copyedit and formatting clean-up (typos, merged words, repeated phrases, spacing/punctuation).

cGPA scale clarification: Explicitly state the maximum cGPA scale used and align all categories accordingly.

Generalizability wording: Keep conclusions appropriately cautious given the convenience sampling and institutional dominance.

Once these minor edits are completed, the manuscript can be accepted.
---

## [Editor Report · Acceptance letter]

PONE-D-25-49996R3

PLOS One

Dear Dr. Alhemedi,

I'm pleased to inform you that your manuscript has been deemed suitable for publication in PLOS One. Congratulations! Your manuscript is now being handed over to our production team.

Kind regards,

on behalf of

Dr. Nour Amin Elsahoryi

Academic Editor

PLOS One